# Ultrathin Electrolyte Membranes with PFSA-Vinylon Intermediate Layers for PEM Fuel Cells

**DOI:** 10.3390/polym12081730

**Published:** 2020-08-03

**Authors:** Jedeok KIM, Kazuya Yamasaki, Hitoshi Ishimoto, Yusuke Takata

**Affiliations:** 1Hydrogen Production Materials Group, Center for Green Research on Energy and Environmental Materials, National Institute for Materials Science (NIMS), 1-1 Namiki, Tsukuba, Ibaraki 305-0044, Japan; 2Functional Clay Materials Group, Research Center for Functional Materials, National Institute for Materials Science (NIMS), 1-1 Namiki, Tsukuba, Ibaraki 305-0044, Japan; 3Engineering Division, Industrial Solutions Company, Panasonic Corporation, 1006 Kadoma, Kadoma City, Osaka 571-8506, Japan; yamasaki.kazuya001@jp.panasonic.com (K.Y.); ishimoto.hitoshi@jp.panasonic.com (H.I.); takata.y@jp.panasonic.com (Y.T.)

**Keywords:** PFSA ionomers, PFSA-vinylon, lamination, THIN membrane, fuel cells

## Abstract

We prepared ultrathin PFSA/PFSA-vinylon/PFSA laminated electrolyte membranes (thickness = 10 μm) for fuel cells without using a reinforcing material. Nafion and Aquivion solutions were used as PFSA polymers. Vinylon was synthesized by formalizing polyvinyl alcohol. From the current-voltage measurements using ultrathin PFSA/PFSA-vinylon/PFSA membranes; the cell resistances are significantly lower than that using a 50 μm Nafion membrane. A high current density was obtained under both low- and high-humidity conditions. Ultrathin PFSA/PFSA-vinylon/PFSA laminated membranes will help to further improve the performance of PEMFCs.

## 1. Introduction

Fuel cells, water electrolysis devices, secondary batteries, redox flow batteries (RFBs), solar cells, etc., are attracting attention as important energy devices for realizing a CO_2_-free society. Research to improve the performance of these devices and systems is actively being conducted. The performance and durability of fuel cells have been improving while their system costs have reduced, and they are being used in mobile and stationary applications. However, in order to be commercialized, the constituent materials must be made cheaper, and the performance and durability must be higher. One way to achieve low cost and high performance is to decrease the thicknesses of the polymer electrolyte membranes. Proton exchange fuel cells utilize a polymer electrolyte composed of perfluorosulfonic acid (PFSA) type ionomers, such as DuPont’s Nafion and, more recently, 3M’s and Solvay’s ionomers [1]. The thickness of the electrolyte membranes has been reduced from 178–25 μm for a Nafion membrane to 20–5 μm for a GORE-SELECT^®^ membrane [2,3]. Thinning the polymer electrolyte membrane can improve the performance by reducing the ohmic voltage drop of the fuel cell and can reduce the cost. However, extremely thin polymer electrolyte membranes are difficult to handle when assembling a fuel cell, and their durabilities are lower. On the other hand, the GORE-SELECT^®^ membrane, which uses expanded polytetrafluoroethylene (ePTFE) as a reinforcing material, is a thin membrane that can be handled reliably and has a high durability, but it is costly [3,4,5]. Therefore, a highly durable thin membrane without reinforcing materials that is easy to handle would be useful as an electrolyte membrane. There has been a report of a Nafion/sulfonated polyimide (SPI)/Nafion composite membrane without a reinforcing membrane with a thickness of 15 μm [6]. However, it is a layered membrane prepared by dipping a hydrocarbon-based SPI membrane, which is an engineering plastic with a thickness of 11 μm, into a Nafion solution. There are no reports of fuel cell evaluation using an ultrathin PFSA membrane of 10 μm or less without a reinforcing material.

In this communication, we prepared an ultrathin laminated membrane of PFSA/PFSA-vinylon/PFSA that could be handled even when the thickness was under 10 μm by inserting a PFSA-vinylon composite layer between PFSA electrolyte membranes. Vinylon has high hydration stability and scale stability, so it is suitable as an additive material for thin membrane formation. In addition, it was determined to have a high current density from fuel cell evaluations.

## 2. Experimental

Nafion (5%, DE520 CS type, equivalent weight (EW) = 1100) and Aquivion solutions (6%, D83-06A, equivalent weight (EW) = 830) were purchased from Fujifilm Wako Pure Chemical Corporation (Osaka, Japan) and Solvay (Tokyo, Japan), respectively. Polyvinyl acetate (PVAc, (C_4_H_6_O_2_)_n_, Mw = 86.09 g/mol, Mw = 100,000) was purchased from Sigma-Aldrich Co., Ltd. (St. Louis, MO, USA). Sodium hydroxide (NaOH), methanol (CH_3_OH), formaldehyde (HCHO, 37%), sodium sulfate (Na_2_SO_4_), hydrogen peroxide (H_2_O_2_), sulfuric acid (H_2_SO_4_), and sodium chloride (NaCl) were purchased from Nacalai Tesque (Kyoto, Japan). A Nafion212 (NR212) membrane was purchased from DuPont (USA). DI water was obtained using a Purelab^®^ Option-R ELGA Labwater (25 °C, 15 Mohm cm).

The synthesis of polyvinyl alcohol (PVA) has been described in a previous paper [7]. The PFSA-PVA mixed solution was prepared using the following method. To obtain a uniform mixture, the amount of PVA added for Nafion was 5 wt % PVA, and 1 wt % PVA was added for Aquivion.

The PFSA/PFSA-PVA/PFSA membrane was prepared as follows. A polyimide sheet (27 cm × 30 cm) was placed on a coater (KP-3000VH, KIPAE CO. Ltd., Suwon, Korea) and coated with an applicator (Tester Sangyo CO. Ltd., Saitama, Japan; 15 cm, gap = 0.1 mm) at room temperature. Before producing the laminated membrane, the membrane thickness was controlled by using a single membrane, and then the laminated membrane was prepared. First, 2 μm PFSA layer membranes were coated and then dried. Then, intermediate layers of 6 μm PFSA-PVA layer membranes were coated and then dried, and finally, a 2 μm PFSA layer membrane was coated and dried. Then the membranes were heated at 60 °C for 3 h, followed by 180 °C for 3 h, to obtain 10 μm PFSA/PFSA-PVA/PFSA laminated membranes.

To form vinylon, the PFSA/PFSA-PVA/PFSA membranes were added to a formalization solution with a mass ratio of H_2_O:H_2_SO_4_:Na_2_SO_4_:CH_2_O of 1.00:0.21:0.20:0.06 at 60 °C for 2 h and were then washed with water to afford PFSA/PFSA-vinylon/PFSA membranes. They were activated with boiling water (2 h), 1M H_2_SO_4_ (80 °C, 2 h), and boiling water (2 h) and dried at room temperature.

The ion exchange capacity (IEC), water-uptake (W.U.), and proton conductivity were measured by using the methods reported in our previous paper [8]. The chemical structure of the sample was determined using attenuated total reflection (ATR) Fourier transform infrared (FTIR) spectroscopy on a Thermo Scientific Nicolet 6700 spectrometer. The PFSA/PFSA-PVA/PFSA membranes used for fuel cell measurements had a thickness of 10 μm, and the Nafion212 membrane used for comparison had a thickness of 50 μm. A catalyst ink was prepared by mixing 50 wt % Pt/C (TEC10E50E, Tanaka Kikinzoku Kogyo K.K.) and 20 wt % Nafion ionomer (DE2020, Fujifilm Wako Pure Chemical Corporation). The mass ratio of the binder and carbon balance (ionomer/carbon) was adjusted to 1.0. The catalyst ink (0.3 mg-Pt cm^−2^) was formed on the gas diffusion layer (GDL) (SGL Group Co. Ltd., Tokyo, Japan) and hot-pressed at 140 °C and 1 MPa for 5 min to fabricate a membrane electrode assembly (MEA). The effective electrode area of the single cell was 4 cm^2^. I–V characteristics were measured at 80 °C and atmospheric pressure under 100% and 35% RH using hydrogen for the anode and air or oxygen for the cathode. The flow rates of the gases were controlled using a mass flow controller (MFC). The amounts of reaction gases were 75% for hydrogen (H_2_) and 55% for oxygen (O_2_).

## 3. Results and Discussion

Figure 1 shows a photograph and a schematic diagram of an ultrathin Nafion/Nafion-vinylon/Nafion membrane, and the chemical structure of the intermediate Nafion-vinylon layer. Although it is difficult to handle the ultrathin PFSA membrane alone (10 μm), ultrathin PFSA/PFSA-vinylon/PFSA membranes with thicknesses of 10 μm or less could be handled. It is thought that the mechanical properties are improved by adding a PFSA-vinylon layer. From a cross-sectional observation of the PFSA/PFSA-vinylon/PFSA membrane by using scanning electron microscopy (SEM), the laminated membranes could not be distinguished. Since the layers were composed of the same components, there may have been no interface. However, a more detailed structural analysis is required.

The existence of vinylon in the ultrathin PFSA/PFSA-vinylon/PFSA membrane was investigated by using FTIR. Figure 2 shows FTIR spectra for Aquivion only and Aquivion/Aquivion-vinylon/Aquivion membranes. In the spectrum of the Aquivion/Aquivion-vinylon/Aquivion membrane, peaks for vinylon were observed at 2918 cm^−1^ (*v*_as_, CH_2_) and 2842 cm^−1^ (*v*_s_, CH_2_) in addition to a peak for Aquivion. From these results, the presence of vinylon in the ultrathin membrane was confirmed.

The conductivities of the laminated ultrathin membranes were measured in relation to the relative humidity at 40 and 80 °C. Figure 3 shows the conductivity characteristics of the Nafion/Nafion-vinylon/Nafion and Aquivion/Aquivion-vinylon/Aquivion membranes. The conductivity of the ultrathin laminated membranes increased with increases in the temperature and the humidity. The conductivity of Aquivion/Aquivion-vinylon/Aquivion was higher than that of Nafion/Nafion-vinylon/Nafion. The IEC of Nafion/Nafion-vinylon/Nafion was 1.12 meq/g, and the W.U. was 50%. The IEC of Aquivion/Aquivion-vinylon/Aquivion was 1.46 meq/g, and the W.U. was 86%. The higher conductivity of the Aquivion-based membrane in comparison to the Nafion-based membrane was due to the high IEC and W.U. Moreover, the conductivity depended on the membrane thickness, meaning that the conductivity decreased with a decrease in the thickness. A Nafion212 membrane with a thickness of 50 μm has been reported to have a conductivity of about 0.1 S/cm at 80 °C under 90% RH [1,8,9].

Recast ultrathin membranes of about 10 μm prepared using Nafion and Aquivion solutions are difficult to handle, and they are easily broken. However, the laminated ultrathin membranes could be handled even with a thickness of 10 μm or less, and the fuel cell performance could be evaluated. Figure 4 shows the results of current-voltage (I–V) and current-resistance (I-R) measurements made using fuel cells with MEAs fabricated with 10 μm Nafion/Nafion-vinylon/Nafion and Aquivion/Aquivion-vinylon/Aquivion membranes. In addition, the characteristics of a Nafion212 membrane with a thickness of 50 μm are shown for comparison. The measurements were performed at a cell temperature of 80 °C under 100% RH and 35% RH conditions. The anode gas was hydrogen, and the cathode gas was air and oxygen.

From Figure 4, the resistances of the fuel cells using the 10 μm laminated ultrathin membranes were significantly lower in comparison to the resistances of the fuel cell using the 50 μm Nafion212 membrane. This is because thinning an electrolyte membrane reduces the cell resistance of a fuel cell. Due to the low resistance, the I–V performances of the fuel cells using the laminated ultrathin membranes were significantly better than those using the Nafion212 membrane. In addition, the I–V performances of the fuel cells using the laminated ultrathin membranes and the Nafion212 membrane were more remarkable when oxygen was used than when air was used as the cathode gas. Furthermore, the Aquivion/Aquivion-vinylon/Aquivion membrane was stable with no increase in the resistance even at a high current density of 3 A/cm^2^. On the other hand, the open-circuit voltage of the fuel cells using the 10 μm laminated ultrathin membranes decreased. This indicates that the gas barrier properties of the laminated ultrathin membranes are lower than those of the Nafion212 membrane, and therefore, it is necessary to improve the gas barrier properties. Table 1 summarizes these characteristics.

## 4. Summary

We developed laminated ultrathin membranes that could be handled even when the thickness was 10 μm or less, by combining a PFSA (Nafion and Aquivion) polymer with vinylon and laminating it with PFSA polymer. From the I–V evaluation of the fuel cells using the ultrathin PFSA/PFSA-vinylon/PFSA membranes, the cell resistances were low in comparison to those using the Nafion212 membrane, and high performances were obtained. Moreover, when oxygen was used as the cathode gas, the performance was remarkably high. We propose that laminating ultrathin PFSA-vinylon membranes with PFSA is a good method for lowering costs, achieving high performance and eliminating the need for a reinforcing material.

## Figures and Tables

**Figure 1 polymers-12-01730-f001:**
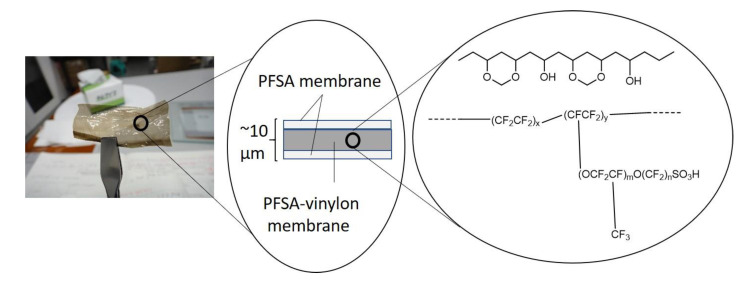
Ultrathin Nafion/Nafion-vinylone/Nafion membrane, schematic diagram, and Nafion-vinylon chemical structure.

**Figure 2 polymers-12-01730-f002:**
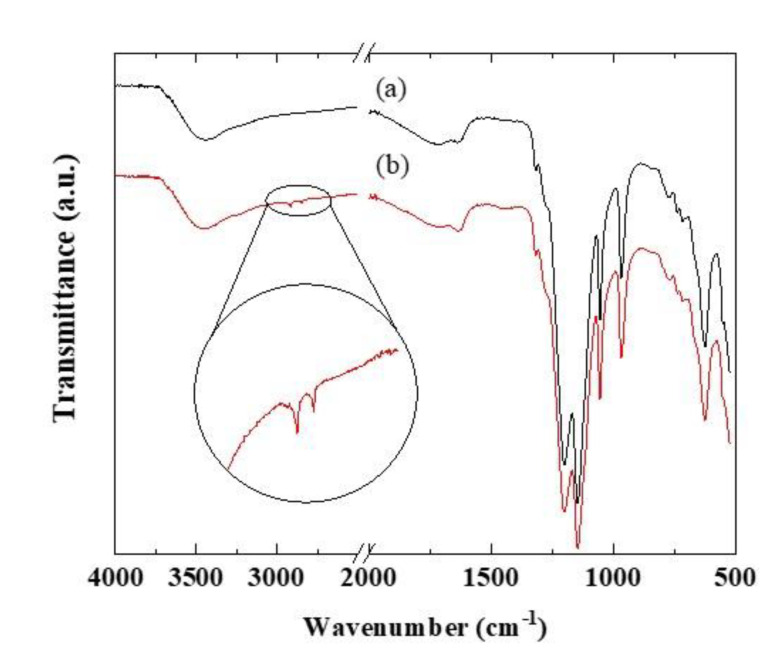
FTIR spectra for (a) Aquivion and (b) an Aquivion/Aquivion-vinylon/Aquivion membrane.

**Figure 3 polymers-12-01730-f003:**
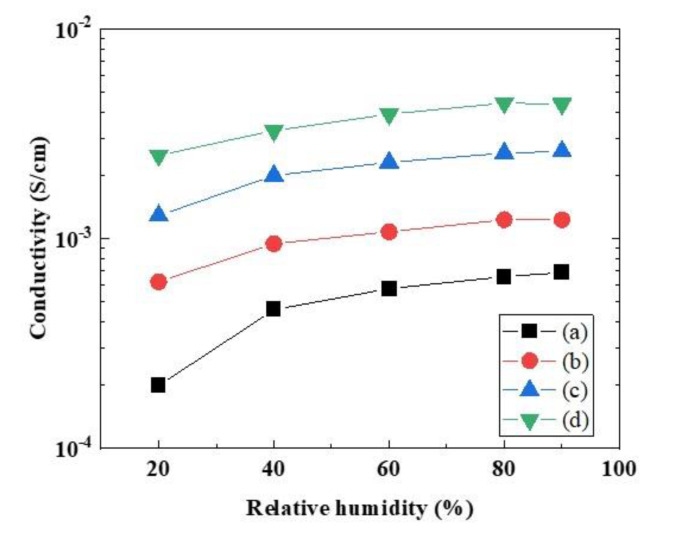
Proton conductivities of Nafion/Nafion-vinylon/Nafion (a) at 40 and (b) 80 °C and Aquivion/Aquivion-vinylon/Aquivion (c) at 40 and (d) 80 °C.

**Figure 4 polymers-12-01730-f004:**
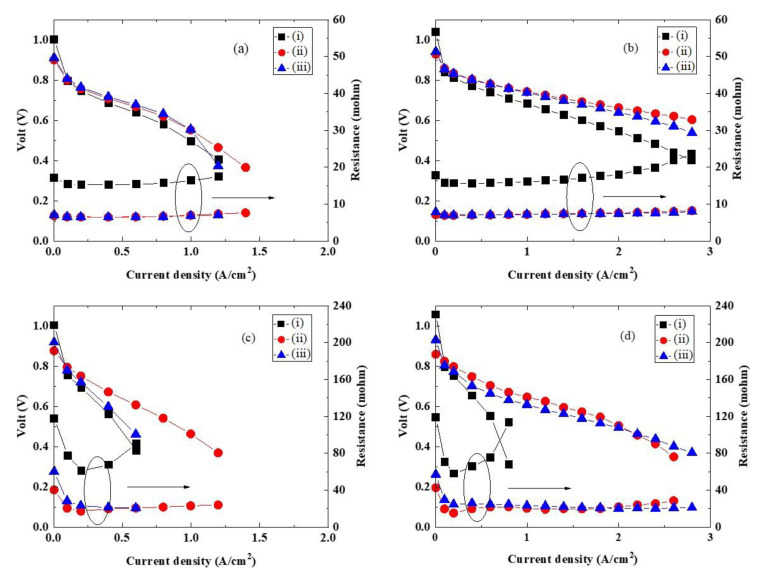
I–V characteristics of (i) Nafion212, (ii) Nafion/Nafion-vinylon/Nafion, and (iii) Aquivion/Aquivion-vinylon/Aquivion at 80 °C in (**a**) air and (**b**) O_2_ under 100% RH and in (**c**) air and (**d**) O_2_ under 35% RH.

**Table 1 polymers-12-01730-t001:** I–V performance summary for the (i) Nafion212, (ii) Nafion/Nafion-vinylon/Nafion, and (iii) Aquivion/Aquivion-vinylon/Aquivion membranes.

Membranes	100% RH				35% RH		
	Air		O_2_		Air		O_2_
	Resistance (mohm) at 0.3 A/cm^2^	Rate (%) of Decreasing Resistance	Resistance (mohm) at 2.5 A/cm^2^	Rate (%) of Decreasing Resistance	Resistance (mohm) at 0.3 A/cm^2^	Rate (%) of Decreasing Resistance	Resistance (mohm) at 2.5 A/cm^2^
(i)	15.6	100	20.8	100	60.8	100	_
(ii)	6.5	45	8.0	38	16.7	27	26.4
(iii)	6.5	45	7.7	37	25.1	41	20.3

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
