# Peer review of "Ultrathin Electrolyte Membranes with PFSA-Vinylon Intermediate Layers for PEM Fuel Cells"

_polymers, 2020, doi:10.3390/polym12081730_

Round 1
Reviewer 1 Report
The manuscript reports a thin PFSA membrane for fuel cell applications. Good conductive performance and cell performance was demonstrated. The manuscript can be reconsidered after addressing the following major issues. 1. The authors should clearly clarify the design principle of the membrane. Why polyvinylon was selected for the design of the membrane? 2. Is polyvinylon stable under strong acidic fuel cell condition? The authors should perform ex-situ experiments to demonstrate this point. 3. The results show that the thin composite membrane has better perofmrance than the thick Nafion 212 membrane, which is not suprsing. The authors should compare the fabricated thin membrane with a commercial membrane of similar thickness. 4. The mechanical perfomrance of the thin membrane could be a possible concern for its applcilation. The authors should add relating experiments to show the applicability of the membrane.Author Response
Answers to Reviewer 1
I would like to thank for your kind reading of our paper. According to the comments, I have revised our manuscript with yellow highlighter. Answers to the reviewer are listed below.
Reviewer 1
The manuscript reports a thin PFSA membrane for fuel cell applications. Good conductive performance and cell performance was demonstrated. The manuscript can be reconsidered after addressing the following major issues. 1. The authors should clearly the design principle of the membrane. Why polyvinylon was selected for the design of the membrane? 2. Is polyvinylon stable under strong acidic fuel cell condition? The authors should perform ex-situ experiments to demonstrate this point. 3. The results show that the thin composite membrane has better performance than the thick Nafion212 membrane, which is not surprising. The authors should compare the fabricated thin membrane with a commercial membrane of similar thickness. 4. The mechanical performance of the thin membrane could be a possible concern for its application. The authors should add relating experiments to show the applicability of the membrane.
Thank you for your comments. I would like to answer your comments.
- The authors should clearly the design principle of the membrane. Why polyvinylon was selected for the design of the membrane?
As you know, PVA has been studied as an additive in fuel cells for DMFC such as Nafion membrane and SPEEK membrane in order to suppress methanol crossover. However, PVA is highly hydrophilic, unstable in hydration conditions, and weak in mechanical properties. However, by converting PVA to vinylon, it is stable under hydration conditions and the mechanical properties are greatly improved. Especially, the mechanical properties are excellent under the thin membrane. We have made it possible to reduce the membrane thickness and cost by making use of the characteristics of vinylon to make a laminated membrane, which is a problem that it is difficult to reduce the membrane thickness (10μm or less) by using the PFSA membrane alone. Also, handling is possible even at 5μm or less. Furthermore, the addition of vinylon improved the fuel cell characteristics instead of deteriorating it, and improved even in a low humidity environment. These facts suggest that vinylon affects not only the mechanical properties but also the water and proton conduction paths of the membrane.
- Is polyvinylon stable under strong acidic fuel cell condition? The authors should perform ex-situ experiments to demonstrate this point.
The produced laminated membrane was activated in boiling water for 2 hours, in 1M sulfuric acid at 80°C for 2 hours, and in boiling water for 2 hours, so it is stable in these environments. However, we also investigated the characteristics of Vinylon only. Vinylon was stable in 1M sulfuric acid at 85°C for 2 hours. Therefore, it is stable with such a strong acid. More details will be reported in the next paper.
- The results show that the thin composite membrane has better performance than the thick Nafion212 membrane, which is not surprising. The authors should compare the fabricated thin membrane with a commercial membrane of similar thickness.
As mentioned in the paper, it is not easy to obtain a 10 μm thin membrane with a fluorine electrolyte solution. In particular, it is very difficult to obtain a large thin membrane for fuel cell evaluation with commercially available fluorine-based solutions. We made it easy with vinylon. Furthermore, even when an insulator called vinylon was added, higher performance was obtained than in previous reports. It is also higher than the performance of the Nafion 211 membrane (25 μm) in Reference 1 below. Moreover, it is higher than the performance of the GORE SELECT membrane (10 μm) of Reference 2. It can be said that the performance of 3A/cm2 or more at 0.6V (Fig. 4b) was amazing.
Ref.1) J. Peron, A. Mani, X. Zhao, D. Edwards, M. Adachi, T. Soboleva, Z. Shi, Z. Xie, T. Navessin, S. Holdcroft, Properties of Nafion® NR-211 membranes for PEMFCs. J. Membrane Science. 2010, 356, 44-51.
Ref.2) B. Kienitz, J.Kolde, S. Priester, C.Baczkowski, M. Crum, Ultra-thin reinforced ionomer membranes to meet next generation fuel cell targets. Electrochem. Soc. Trans. 2011, 41(1) 1521-1530.
- The mechanical performance of the thin membrane could be a possible concern for its application. The authors should add relating experiments to show the applicability of the membrane.
Yes. That's right. Mechanical properties are especially required when thinning the membrane. The laminated membrane is very flat on the polyimide (PI) substrate, but when peeled off from the PI film, it is difficult to make it flat due to the stress, and it is difficult to evaluate the mechanical properties. However, as can be seen from the OCV results of the fuel cell (Fig. 4), the mechanical properties need further improvement. Research on more reinforced thin membranes is under consideration and will be reported in the next paper.

Reviewer 2 Report
The article presented refers to the development of ultra-thin laminated PFSA (commercial Nafion and Aquivion)/vinylon-PFSA intermediate/PFSA membranes for PEMFCs without reinforcement and with low cell resistances.The results are interesting especially for the construction method and for the performances achieved.
Only small considerations: Page 1 line 36 correct polytetraethylene with polytetrafluoroethylene. Page 2 line 49 add the equivalent weight (EW) of the two ionomers in solutions, 1100 for the Nafion and 830 for the Aquivion. In fact in line 120 the high IEC value is also given by the lowest EW for Aquivion. As for the references, i suggest to add others to reinforce this communication.
Author Response
Answers to Reviewer 2
I would like to thank for your kind reading of our paper. According to the comments, I have revised our manuscript with yellow highlighter. Answers to the reviewer are listed below.
Reviewer 2
The article presented refers to the development of ultra-thin laminated PFSA (commercial Nafion and Aquivion)/vinylon-PFSA intermediate/PFSA membranes for PEMFCs without reinforcement and with low cell resistances.
The results are interesting especially for the construction method and for the performances achieved.
Only small consideration: Page 1 line 36 correct polytetraethylene with polytetrafluoroethylene.
Page 2 line 49 add the equivalent weight (EW) of the two ionomers in solutions, 1100 for the Nafion and 830 for the Aquivion. In fact in line 120 the high IEC value is also given by the lowest EW for Aquivion. As for the references, i suggest to add others to reinforce this communication.
Thank you for your comments.
Revised the paper based on the comments and added 2 references for the reinforced membranes.
Please check the paper.
